# Photogenerated Carrier-Assisted Electrocatalysts for Efficient Water Splitting

**Xiang Li [1], Xueyan Zheng [1], Yanzhong Zhen [1,\*] and Yucang Liang [2,\*]**

1    Yan'an Key Laboratory of Green Hydrogen Energy and Biomass Catalytic Conversion,
     School of Chemistry & Chemical Engineering, Yan'an University, Yan'an 716000, China
2    Institut für Anorganische Chemie, Eberhard Karls Universität Tübingen, Auf der Morgenstelle 18,
     72076 Tübingen, Germany
*    Correspondence: zhenyanzhong@yau.edu.cn (Y.Z.); yucang.liang@uni-tuebingen.de (Y.L.)

**Abstract:** Electrocatalysts are the core component of electrocatalytic water splitting for improving its overall energy conversion efficiency and reducing the energy input. At present, the design of efficient electrocatalysts mainly focuses on optimizing their electronic structure and local reaction microenvironment to improve the adsorption of reaction intermediates. Although many effective strategies (such as heteroatom doping, vacancy, heterojunction construction, strain engineering, and phase transformation) have been developed, the improvement in catalytic activity has been very limited. Hence, the development of innovative strategies to enhance the optimization of photoelectroactivity is desirable. Inspired by the strategy of applying a potential field to reduce carrier radiation recombination in traditional photoelectrocatalysis, photogenerated carrier-assisted electrocatalysis, based on the synergy effect of light and electric energy, provides a new strategy to enhance the intrinsic activity of water splitting. The essence of the photo-assisted strategy can be attributed to the injection of hot carriers and photogenerated electron–hole pairs or the accelerated reaction kinetics caused by local temperature rises. The photogenerated carrier-assisted strategy has received wide attention due to its simplicity and efficiency. In this review, we focus on the recent advances in photogenerated carrier-assisted strategies (PCAS) for enhancing the performance of HER, OER, and overall water splitting. The possible mechanisms are addressed and the basic composition and latest progress in photo-assisted electrocatalysts using PCAS are summarized. Finally, the challenges and development prospects of PCAS will be detailed.

**Keywords:** photogenerated carrier-assisted electrocatalysis; HER; OER; overall water splitting; catalytic mechanism

## 1. Introduction

Electrochemical water splitting technology driven by renewable energy for hydrogen production, due to its sustainability and environmental friendliness, has been considered to be an effective strategy to solve the problem of resource shortages caused by the rapid consumption of fossil fuel energy and realize green energy conversion in the future [1–3]. However, the slow dynamics of the two half-cell reactions in the hydrogen evolution reaction (HER) and the oxygen evolution reaction (OER) results in an inferior energy conversion efficiency [4]. To meet the expected demand for this industrially practical application, efficient electrocatalysts are essential for reducing the activation energy barrier of water splitting and accelerating two terminal reactions [5,6]. Owing to the complex multiphase physiochemical environment and multi-electron transfer involved in the process of water splitting, an ideal electrocatalyst should possess high conductivity, a suitable adsorption and desorption capacity, and high structural stability [7]. Currently, platinum group metals have an onset potential approaching the theoretical thermodynamic potential, and are regarded as the optimum electrocatalysts for water splitting. However, the high price

and scarcity of precious metals limit their large-scale application [6,8–10]. In order to improve the cost-effectiveness, it is imperative to develop a robust substitute for noble metal catalysts composed of earth-abundant elements [1]. Transition-metal-based materials, with their advantages of rich reserves, good electronic conductivity, multiple redox valences, and unsaturated transition metal sites, have been widely investigated for overall water splitting and have achieved remarkable results [11]. However, transition metal electrocatalysts are still limited by inherent performance bottlenecks; the improvement in the activity and durability of pristine electrocatalytic materials is still the emphasis of current research [12,13].

The deepening of theoretical research and the progress of advanced characterization techniques help us to comprehensively understand the relationship between the structure/elemental properties of the catalysts and their reaction mechanisms and catalytic activity, which can provide guidelines for the rational design of electrocatalysts [14,15]. In general, optimizing the adsorption energy of intermediates plays a decisive role in improving the catalytic performance, and the regulation of the electronic structure of the active sites can effectively ameliorate the adsorption free energy of the catalyst for intermediate products [16,17]. Heteroatom doping and the surface vacancy can change the local charge distribution and produce defects, and studies have shown that through their integration, a more optimized electronic state can also be achieved cooperatively [17–19]. The construction of a heterojunction can induce charge rearrangement at the interface of the heterojunction and modify the properties of active sites, and recent studies have also identified the key role of heterogeneous catalysts in promoting the lattice oxygen oxidation mechanism (LOM) [20–22]. Strain engineering can directly affect the atomic arrangement and lattice structure of electrocatalysts; introducing a small strain can obviously cause the displacement of the d band center [11,23]. Phase transitions can adjust the surface adsorption properties and charge state, which will result in a better conductivity and a higher surface activity [24]. All these effective strategies can regulate the electronic structure. In addition to rational design of electrocatalysts from the above points of view, it is also particularly important to explore innovative strategies to improve the catalytic activity. For instance, the studies have shown that pre-catalysts can achieve a dynamic activation process in electrochemical water splitting by controlling the dissolution, the leaching of intrinsic elements, and surface reconstruction [25–27].

In recent years, multi-field coupling technology has been widely applied in the catalysis field. Introducing some external fields effects can optimize the catalytic process, and the reasonable utilization of resources can also be realized through the interconnection gain between energy fields [28,29]. Electrocatalytic and photocatalytic water splitting are essentially the same, both of which are redox reactions involving an electron transfer process [30]. Benefitting from this, the integration of light field and electric field has become possible and has been practiced in traditional photoelectrochemical (PEC) systems. In general, a PEC system is composed of a photoanode and a photocathode to independently drive the corresponding oxidation and reduction reactions, and an appropriate photoelectrode must have a matching band gap and energy band position [31,32]. In photoelectrocatalytic water splitting processes, light is the main driving force and the introduction of a small amount of electrical energy can help the reaction system to regulate the radiation recombination of some electron hole pairs and accelerate the separation of carriers excited by semiconductor photoelectrodes under sunlight [33,34]. Although the mechanism of photoelectrocatalytic water splitting has been relatively well-studied and a large number of photoelectric materials have been developed [35–38], the overall solar to hydrogen (STH) conversion efficiency is not still satisfactory. Inspired by electro-assisted photocatalysis strategies, photogenerated carrier-assisted electrocatalysis is considered as an innovative and feasible way to improve the overall catalytic performance. When introducing the light field, the electronic structure and surface reaction microenvironment of a specific electrocatalyst will be affected and thereby the internal catalytic pathway and performance will significantly change [39,40]. So far, the photothermal effect and the local surface plasmon

resonance (LSPR) effect of noble metal nanoparticles and photogenerated carriers have been demonstrated to be effective strategies to assist in enhancing the activity of electrocatalytic HER or OER [41,42]. The photothermal effect generates a local high temperature to enhance the thermodynamic and kinetic processes of electrocatalytic reactions through the collision of photoexcited phonons, which is applicable to most electrocatalysts, but the stability of the catalyst structure at high temperature still needs to be considered [39,43]. Hot carriers excited by plasma radiation undergoing non-radiative decay in the LSPR effect have a higher energy than those directly excited by light and are hence the essence behind the enhancement in electrocatalytic activity, but the inevitable local heating of the surrounding environment caused by the internal relaxation of the high energy carrier makes distinguishing it from the photothermal effect challenging. In addition, high cost precious metals further limit the wide application of this strategy and transition metal/semiconducting metal oxide bifunctional catalysts are becoming more attractive [42,44]. In comparison, the photogenerated carrier-assisted strategy shows broader research prospects. Photogenerated carriers produced by photo-responsive electrocatalysts absorbing light energy can directly participate in the redox reaction and reduce the potential reaction barrier. The whole process is simple and efficient and can be realized by embedding the photoactive components [45,46]. However, there are few studies on photogenerated carrier-assisted electrocatalysis at present, and the specific enhancement steps are still unclear [47]. It is necessary to systematically summarize this field to promote the rational design and development of photogenerated carrier-assisted electrocatalysts.

In this review, we first report the innovative strategy of photogenerated carrier-assisted electrocatalysis and elucidate their possible mechanism for the enhancement in performance by summarizing the reaction mechanism of HER and OER. Then, recent advances in photogenerated carrier-enhanced electrocatalytic HER, OER, and overall water splitting are introduced, and their design and fundamental principle are briefly described. Finally, the pivotal issues and challenges of current photoelectric integrated electrocatalysts are addressed, and the future research and applications will be envisioned. The purpose of this review is to help readers understand the application of photogenerated carrier-assisted electrocatalysis and provide effective guidance to rationally design photo-responsive electrocatalysts.

## 2. Fundamental Principle of Photo-Assisted Electrocatalysis Strategies

Electrochemical water splitting consists of a cathode HER and an anode OER, both of which involve multi-electron transfer and the formation of intermediates. Understanding the reaction path is beneficial to the deep understanding of the photogenerated carrier-assisted electrochemical splitting mechanism. Taking an alkaline HER system as an example, due to the low proton concentration, a $H_2O$ molecule directly participates in the reaction and is decomposed into adsorbed hydrogen ($H_{ads}$) and hydroxide ions through the electron transfer process under the action of catalytic sites (Volmer step) [48]. Then, the $H_{ads}$ combines with $H_2O$ (Heyrovsky step) or another $H_{ads}$ under high $H_{ads}$ coverage (Tafel step) to generate $H_2$ [7]. The OER mechanism is mainly composed of a series of proton-coupled electron transfer (PCET) processes based on catalytic active centers, involving the sequential and step-by-step formation of intermediates (OH*, O*, and OOH*) [49,50]. In general, the OER activity is closely related to the adsorption energies of these intermediates. Photogenerated carrier-assisted electrochemical water splitting combines photocatalysis and electrocatalysis, its essence is the redistribution of photogenerated carriers. Compared to the traditional PEC system, the difference is that electric energy is the main driving force of the catalytic reaction [30]. When exposed to light, the photoactive components in the electrocatalyst excite the carriers, which subsequently migrate to participate in the decomposition of water. Photogenerated electrons can effectively diffuse from the conduction band to the electrocatalyst under the action of an external electric field and directly participate in reduction reactions, while reducing the energy barrier required for HER and compensating for power consumption [46,51]. The photogenerated holes in the valence band with strong oxidizability can directly react with the OH$^-$ adsorbed

on the electrode surface to generate OH*. The generated OH* and photogenerated holes subsequently participate in the oxidation of OH- and other intermediates to produce $O_2$ (Figure 1) [52,53]. Simultaneously, the formation of high-valence active sites can be promoted under the oxidation of holes. For instance, Alberto Naldoni et al. investigated a hematite (a-$Fe_2O_3$) photoanode covered with a nickel hydroxide electrocatalyst and confirmed that the nickel hydroxide was completely oxidized to nickel oxyhydroxide by photogenerated holes. Nickel reached a higher oxidation state ($Ni^{IV}$), which is beneficial to the electrochemical oxidation process [54]. With the help of traditional PEC theory, the photogenerated carriers excited by photo-responsive electrocatalysts can be effectively separated under an applied potential, which means that more photogenerated carriers can be injected into the electrocatalytic redox reaction and enable the photogenerated carrier-assisted electrocatalytic water splitting to be more efficient [34]. Therefore, the efficiency of photogenerated carrier-assisted electrocatalysis depends on the effective separation of photogenerated carriers [55]. At the same time, the inherent electrocatalytic activity of photoelectric integrated catalysts under non-optical conditions cannot be ignored. The development and rational design of high-efficiency photoelectric integrated catalysts are crucial for photogenerated carrier-assisted electrocatalysis.

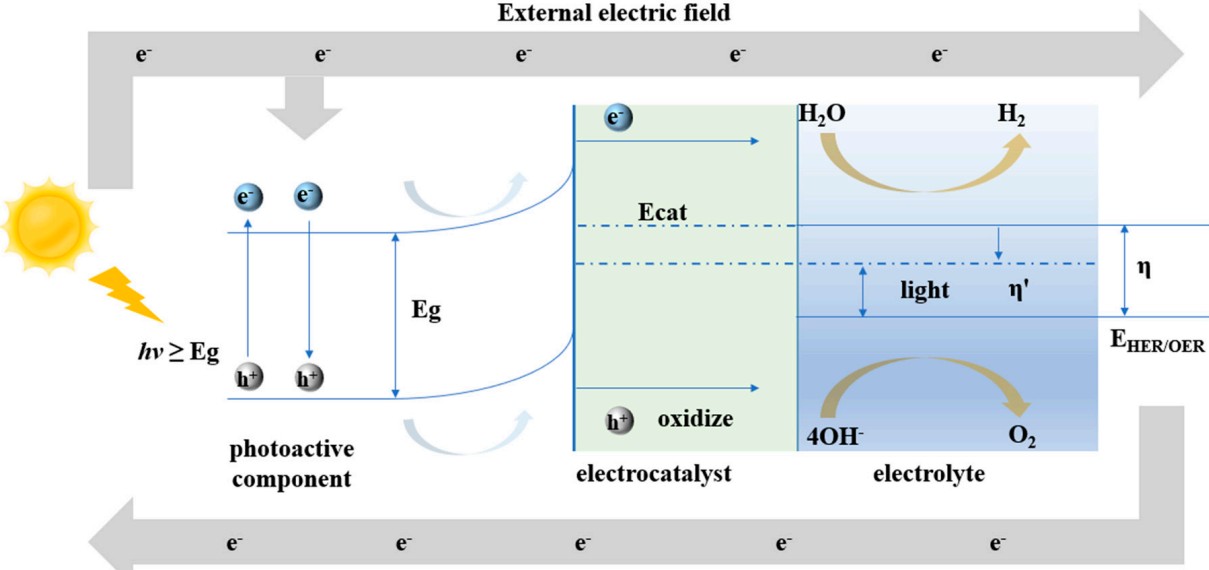

**Figure 1.** The working principle of a photogenerated carrier—assisted electrocatalyst.

### 3. Photo-Electro Integrated Catalysts

In a photoelectrochemical system, the significant recombination of charge carriers, the high overpotential of the interface charge transfer, and the slow reaction kinetics lead to a low overall efficiency [56]. The introduction of cocatalysts can lead to them forming an interface with the semiconductor photoelectrode, capturing the photogenerated carriers, and accelerating their separation, while simultaneously providing surface active sites for the redox reaction to catalyze the reaction of the photogenerated carriers with intermediate ions [57]. Electrocatalysts are well known to have the electrochemical activity of reducing the overpotentials of the HER and the OER, which also exhibits their potential to be used as cocatalysts for photoactive substrates to accelerate the photoelectrochemical kinetics [58,59]. The semiconductor photoelectrode and the introduced electrocatalyst composite can be used as a photoelectric integrated catalyst in the traditional PEC system. In the composite electrode, the electrocatalyst does not play the role of a light absorber. On the contrary, in a photogenerated carrier-assisted electrocatalysis system, the design of photoelectric integrated catalysts can be realized by embedding photoactive components such as semiconductors and quantum dots. The photoactive component acts as the light absorption material, and the electrocatalyst provides the redox reaction active sites. This

reflects the broad prospects of the synergistic effect of light energy and electric energy for electrochemical gain. However, at present, there are still few reports on photogenerated carrier-assisted photoelectric integrated catalysts for electrochemical water splitting. It is necessary to make a systematic summary to promote the development and reasonable design of high-efficiency photoelectric integrated catalysts.

### 3.1. Photogenerated Carrier-Assisted Electrocatalysts for HER

Single-component photoelectric integrated catalysts can naturally couple the electrocatalytic activity with the photoexcitation ability without introducing a photoactive component or an electrocatalytic promoter [60]. At the same time, they also have a strong photoelectric interaction and an efficient charge carrier transfer process, which are very attractive in the photoelectric coupling field [46]. Hao et al. prepared a photo-responsive $Ni_3(VO_4)_2$ electrocatalyst with a sea-urchin-like shape (Figure 2a,d). Single component $Ni_3(VO_4)_2$ showed an enhanced HER performance under light. The unique sea-urchin-like structure was conducive to the exposure of surface active sites and the effective extraction of photogenerated carriers [61]. Rhenium disulfide ($ReS_2$) has a weak interlayer coupling and larger interlayer spacing caused by Peierls distortion, which gives it excellent photoelectric properties and exposes more edge active sites and proton permeation channels [60,62,63]. Zeng et al. found that few-layer $ReS_2$ with the advantages of both light collection and proton reduction kinetics can be used as a new single-component platform for photo-assisted electrocatalytic HER (Figure 2b,e). Due to the guidance of an external electric field and highly active sites, high-energy electrons can be effectively injected and the Fermi energy level of proton reduction can be improved, thus achieving excellent HER performance [60]. Inspired, Xu et al. constructed a $ReS_2/Ni_3S_2$ p-n heterojunction on Ni foam in situ. Benefiting from the strong coupling cooperation between $ReS_2$ and $Ni_3S_2$ heterointerfaces, effective separation of photogenerated carriers can be achieved, and the obtained electrode exhibited an obviously enhanced HER activity (Figure 2c,f) [64].

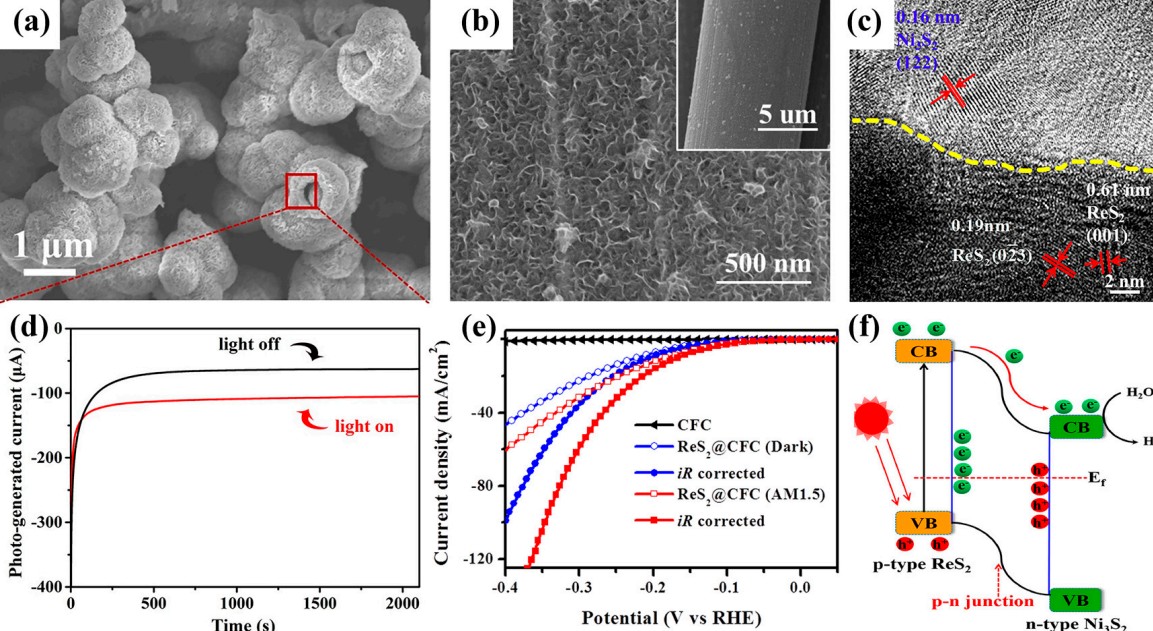

**Figure 2.** (**a**) SEM images of $Ni_3(VO_4)_2$ and (**d**) time—dependent photocurrent responses curve under open circuit potential with/without visible light irradiation. Adapted with permission from ref. [61]. Copyright 2017, Royal Society of Chemistry. (**b**) SEM image and (**e**) LSV curves of $ReS_2$@CFC. Adapted with permission from ref. [60]. Copyright 2018, Elsevier. (**c**) HRTEM image of $ReS_2/Ni_3S_2$ and (**f**) schematic diagram of photogenerated electron transfer path on $ReS_2/Ni_3S_2$ p-n heterojunction under light. Adapted with permission from ref. [64]. Copyright 2019, American Chemical Society.

Quantum dots (QDs), as a rising material in the photocatalytic field, have received much attention for their unique photophysical and photochemical properties. According to the quantum confinement effect, a controllable adjustment of the band gap can be achieved by adjusting the size of the QDs, thus enhancing the optical absorption ability [65]. At the same time, the small particle size is also conducive to the effective separation of charge in QDs for a short transmission distance, while the small particle size endows a larger specific surface area to expose more active sites [66–68]. It is possible to achieve a more effective optical gain by introducing QDs as photoactive components into an electrocatalytic system. Mao et al. successfully constructed dense bannite $Cu_5FeS_4$ QDs on the surface of Ni foam for photo-assisted electrocatalytic HER, and their universal applicability in different electrolytes was verified (Figure 3a). The transient photovoltage (TPV) test indicated the effective extraction of charge in $Cu_5FeS_4$ QDs, which is conducive to the accumulation of surface charge. Combined with DFT theoretical calculations, it was proven that photo-generated electrons can reduce the H* adsorption free energy and optimize the electronic structure of the original electrocatalyst (Figure 3b) [46]. Hao et al. used CdSe QDs as sensitizers and successfully designed and prepared a CdSe QDs/$WS_2$ composite material for HER in a neutral electrolyte solution. Zero-dimensional (0D) and two-dimensional (2D) interfaces were beneficial to the rapid transfer of electrons. The enhanced hydrogen evolution performance under light could be attributed to the transfer of photo-induced electrons in CdSe towards the active sites of $WS_2$ nanosheets (Figure 3c) [69]. In addition, the integration of semiconductor photoactive components and electrochemical active components also shows good prospects in photo-assisted electrochemical systems. Owing to a high conductivity and a tunable surface terminal, 2D $Ti_3C_2T_x$ (MXene) is considered as a promising electrocatalyst. Meanwhile, MXene can be also used as a cocatalyst to accelerate charge separation and transfer in photocatalysis [70,71]. Hao et al. successfully constructed a new type of p-n tungsten oxide homojunction by tuning the phosphorus doping and oxygen vacancies [72]. Based on the synergy of MXene, HER enhanced by visible and near-infrared light can be performed. Under light irradiation, the photogenerated electrons tend to transfer to the n-type $WO_3$ part to participate in the reduction reaction, and the photogenerated holes face the p-type part with a high valence band position. At the same time, the hole trapping ability of MXene and the internal electric field formed by the p-n junction are also conducive to the separation of charges, and thereby greatly improve the photo-assisted efficiency of MXene@P-$WO_3$ (Figure 3d). Similarly, MXene-modified phosphorus-doped $TiO_2$ for photo-assisted electrocatalytic HER has also been reported (Figure 3e). The in situ growth of P-doped $TiO_2$ greatly increased the specific surface area of the composite, and the increased charge density at the edge of the valence band indicated a rapid charge transfer [73]. Additionally, based on the light adsorption ability of $TiO_2$, Ru species supported on MOF-derived N-doped $TiO_2$/C (TC) hybrids were successfully prepared and used as an efficient electrocatalytic–photocatalytic HER catalyst [74]. The resultant high HER performance was attributed to its larger specific surface area and benefits from synergistic coupling of Ru NPs and Ru single atoms (SAs) (Figure 2f). Finally, the main catalysts and experimental conditions and activities of photogenerated carrier-assisted electrocatalysts for HER are listed in Table 1.

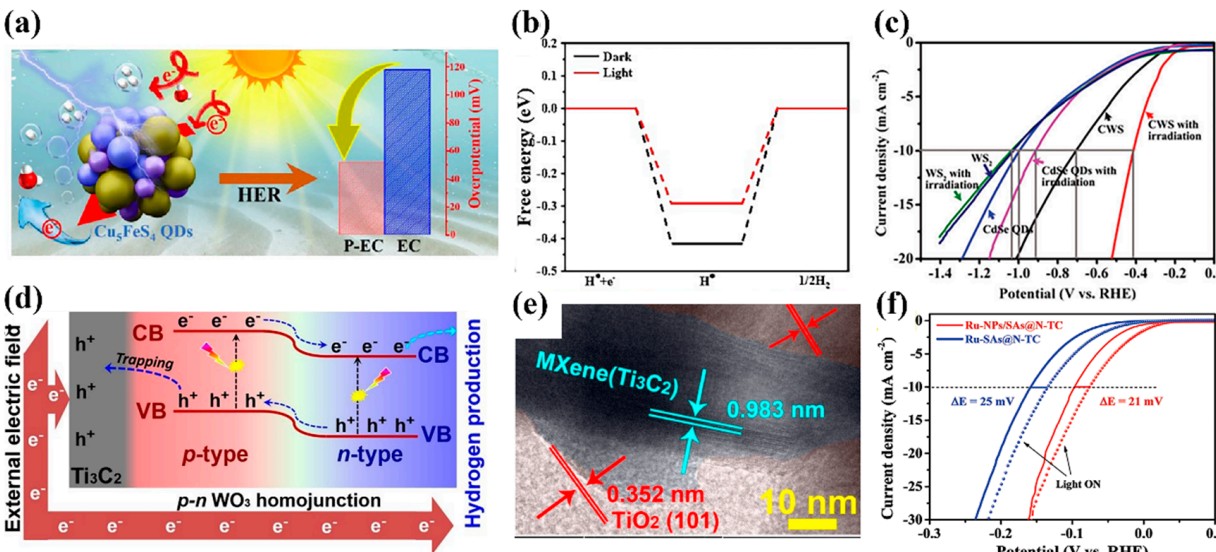

**Figure 3.** (**a**) Schematic diagram of the photo-assisted hydrogen evolution process for $Cu_5FeS_4$ QDs and (**b**) H* adsorption free energy of $Cu_5FeS_4$ QDs in dark and light conditions, respectively. Adapted with permission from ref. [46]. Copyright 2023, Royal Society of Chemistry. (**c**) LSV curves of CdSe QDs/$WS_2$. Adapted with permission from ref. [69]. Copyright 2018, Royal Society of Chemistry. (**d**) Schematic diagram of the charge transfer process in a p-n $WO_3$ homojunction under light irradiation. Adapted with permission from ref. [72]. Copyright 2019, Royal Society of Chemistry. (**e**) HRTEM image of P-$TiO_2$@MXene. Adapted with permission from ref. [73]. Copyright 2021, Elsevier. (**f**) LSV curves of Ru-NPs/SAs@N-TC and Ru-SAs@N-TC. Adapted with permission from ref. [74]. Copyright 2020, Wiley—VCH.

**Table 1.** Main catalysts and experimental conditions and activities of photogenerated carrier-assisted electrocatalysts for HER.

| Catalyst | Electrolyte | Performance Comparison, Light (Dark) | | Photocurrent Density | Ref. |
|---|---|---|---|---|---|
| | | Overpotential /mV | Tafel Slope /mV dec$^{-1}$ | | |
| $Ni_3(VO_4)_2$ | 0.5 M $H_2SO_4$ | 90(122), 10 mA cm$^{-2}$ | 50(82) | NA | [61] |
| $ReS_2$/carbon fiber clothes | 0.5 M $H_2SO_4$ | 167(206), 10 mA cm$^{-2}$ | 80(83) | NA | [60] |
| $ReS_2$/$Ni_3S_2$ | 1.0 M KOH | 106(168), 10 mA cm$^{-2}$ | 111(118) | 0.058 mA.cm$^{-2}$ at $-0.15$ V vs. RHE | [64] |
| $Cu_5FeS_4$ QDs | 1.0 M KOH | 52(118), 10 mA cm$^{-2}$ | 133(143) | NA | [46] |
| CdSe QDs/$WS_2$ | 0.5 M $Na_2SO_4$ | 400(1030), 10 mA cm$^{-2}$ | 56(132) | NA | [69] |
| MXene@P-$WO_3$ | 1.0 M KOH | 44(162), 10 mA cm$^{-2}$ | 41(102) | 200 µAcm$^{-2}$ | [72] |
| MXene@P-$TiO_2$ | 1.0 M KOH | 97(138), 10 mA cm$^{-2}$ | 48.4(73.5) | NA | [73] |
| Ru@N-$TiO_2$/C | 1.0 M KOH | 76(97), 10 mA cm$^{-2}$ | NA | NA | [74] |

### 3.2. Photogenerated Carrier-Assisted Electrocatalysts for OER

OER is a four-electron transfer process, and the slow reaction kinetics greatly limit the overall water cracking efficiency [75]. Therefore, it is necessary to develop highly active oxygen evolution catalysts to reduce the kinetic barrier of OER. At present, noble-metal-based materials such as Ir and Ru still represent the most advanced oxygen evolution catalysts. However, due to their high cost and scarcity, reducing the amount of precious metal loading while improving catalytic performance have become the main problems that must be solved [76]. Liu et al. prepared a $IrO_x$@$In_2O_3$ composite material by the solvothermal method; the composite of $In_2O_3$ could reduce the loading of precious metal $IrO_x$, and the heterojunction exhibited an enhanced OER performance. When a weak LED beam was introduced, the OER performance could be further improved. By amplifying

the LSV curves, Liu et al. also discriminated the photocatalysis-dominated electro-assisted region and electrocatalysis-dominated photo-assisted region during the photogenerated carrier-assisted OER process, which provided effective insights for photogenerated carrier-assisted electrocatalysis strategies (Figure 4a–c) [47]. In addition to precious-metal-based materials, the development of efficient OER electrocatalysts based on non-noble metals is also very important. Transition metal sulfides (TMDs) are considered as promising catalysts for OER due to their unique electronic structure and excellent catalytic activity. The design of photogenerated carrier-assisted electrocatalysts based on TMDs for enhancing OER is equally promising. As a typical TMD, $MoS_2$, with a unique layered structure and a high number of active sites exposed on the edge, has been widely studied as an OER electrocatalyst. Moreover, due to its appropriate band gap structure, $MoS_2$ has also been widely applied as a photocatalyst [77]. Qi et al. introduced black phosphorus quantum dots (BP QDs) into $MoS_2$ nanosheets by the liquid phase stripping method, and the constructed BP QDs/$MoS_2$ heterojunction could realize a photo-enhanced OER performance [78]. The wide absorption band of BP QDs and the rapid electron transfer between 0D/2D heterojunctions results in more photogenerated holes being trapped. In addition, the large specific surface area of BP QDs was conducive to the reaction of photogenerated holes and electrolyte $OH^-$ ions (Figure 4d).

Du et al. reported an advanced CoFe Prussian blue analog (PBA)/$CoS_2$ hybrid material for photogenerated carrier-assisted electrocatalytic OER. As a class of MOF-derived materials based on cyanide, PBAs possess high potential, such as an open skeleton structure, an adjustable composition, and a high specific surface area, which has been widely studied in the field of electrochemistry energy conversion [79,80]. Not only that, its photocatalytic activity in water oxidation has also been confirmed [81]. Based on the synergistic effect between CoFe PBA and $CoS_2$ TMD, the OER performance could be optimized under light irradiation. The enhanced OER activity was dependent on the rapid charge transfer from CoFe PBA to $CoS_2$ TMD, resulting in the enrichment of photogenerated holes on the surface of CoFe PBA, which is greatly beneficial to the promotion of the water oxidation process (Figure 4e,f) [82].

With the help of in situ characterization techniques, transition metal oxyhydroxides have been identified to be the most active for electrocatalytic OER [83]. Hu et al. synthesized a uniform FeOOH nanotube-loaded carbon cloth by the template electroetching strategy and used it as a photo-responsive electrocatalyst. Based on the high OER activity of FeOOH, charge carrier injection further reduced the OER overpotential (Figure 5a–c) [84]. In addition to designing photoelectric integrated catalysts, highly active electrocatalysts are also vital for the modification of photoactive species to enhance the efficiency of photogenerated carrier-assisted electrocatalysis. Lu et al. prepared a p-n heterojunction structured material: p-n $WO_3$/$SnSe_2$ [85]. The p-n heterojunction between $WO_3$ and $SnSe_2$, with a small bandgap, can improve the high carrier recombination in the original $WO_3$ while enhancing its optical absorption ability. The p-n $WO_3$/$SnSe_2$ heterojunction was used to absorb light and generated high energy carriers. Coupling carbon nanotubes (CNTs) with CoFe LDH as the electroactive component leads to the construction of a successful photoelectric integrated catalyst. The coexistence of CNTs and CoFe LDH efficiently separates the photogenerated carriers, and thereby greatly enhances the photo-assisted electrocatalytic efficiency (Figure 5d–f).

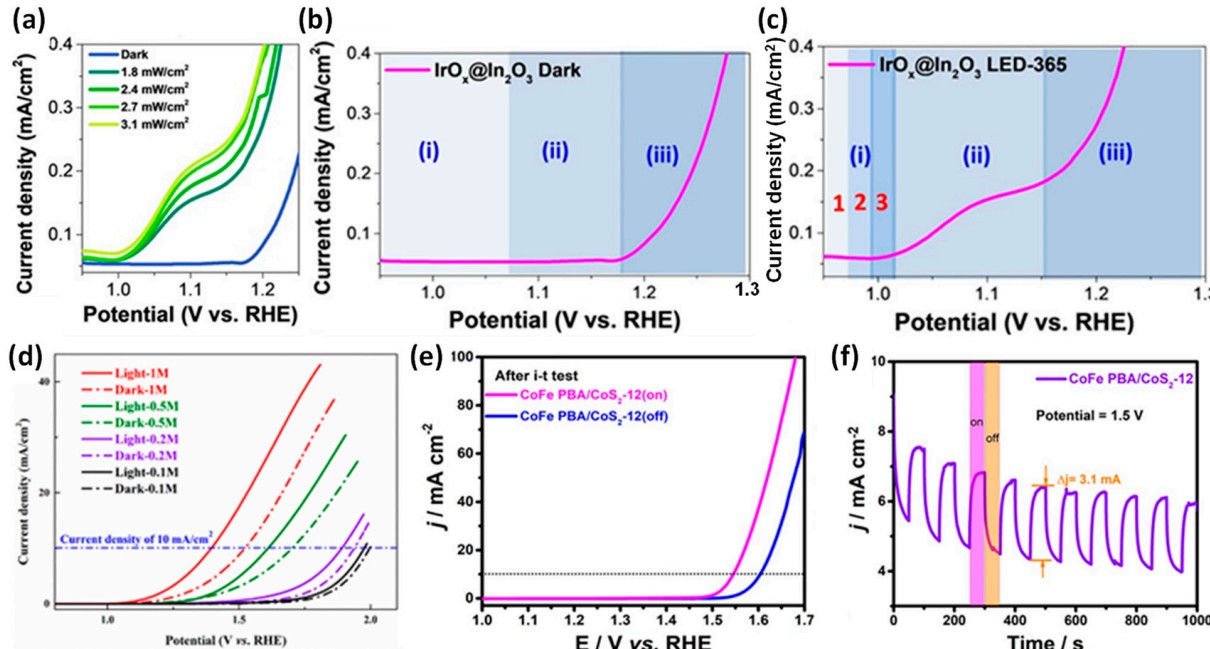

**Figure 4.** (**a**) The enlarged LSV curves of IrOx@In$_2$O$_3$ in the dark and under LED—365 illumination. OER mechanistic analysis of IrOx@In$_2$O$_3$ (**b**) in the dark and (**c**) under LED—365 illumination. Adapted with permission from ref. [47]. Copyright 2021, Wiley-VCH. (**d**) LSV curves of BP QDs/MoS$_2$ in different KOH concentrations under light and dark conditions. Adapted with permission from ref. [78]. Copyright 2021, Elsevier. (**e**) LSV curves of CoFe PBA/CoS$_2$-12 under dark and light irradiation and (**f**) transient photocurrent test of CoFe PBA/CoS$_2$-12 at 1.5 V. Adapted with permission from ref. [82]. Copyright 2019, Royal Society of Chemistry.

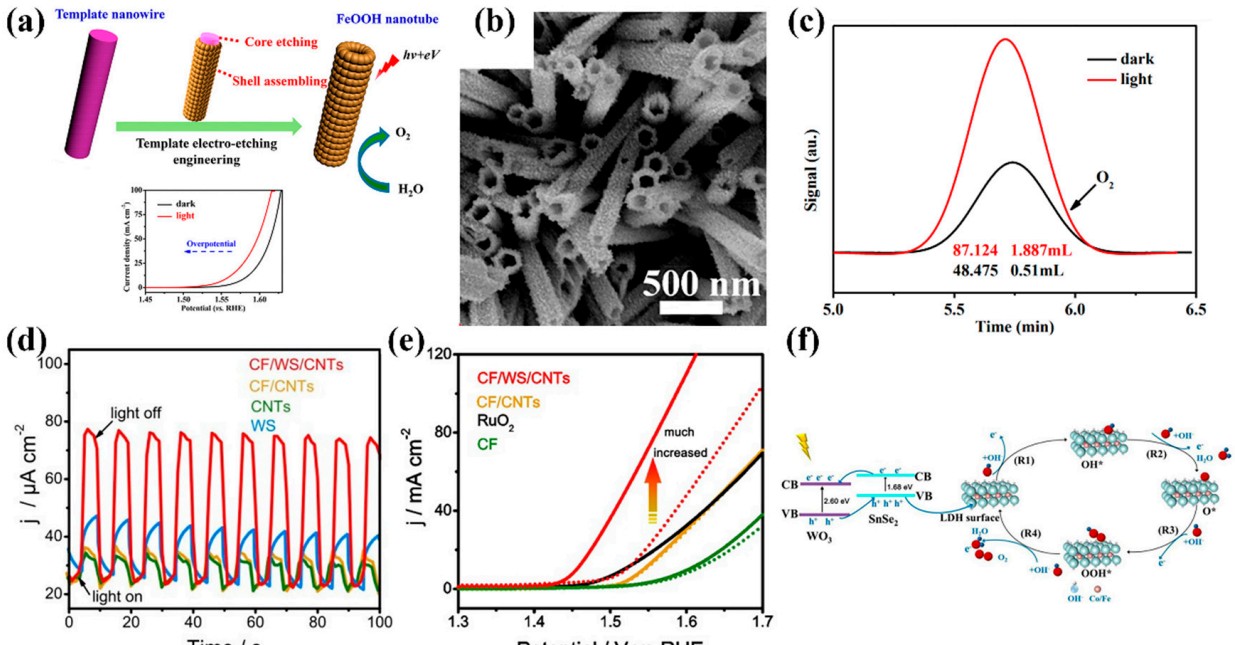

**Figure 5.** (**a**) Schematic diagram of the preparation process for FeOOH and the photoactive electrocatalytic OER curves, (**b**) SEM image of FeOOH nanotubes, and (**c**) the generated O$_2$ content of FeOOH under dark and light conditions. Adapted with permission from ref. [84]. Copyright 2018, American Chemical Society. (**d**) Transient photocurrent density of CF/WS/CNTs, (**e**) LSV curves of CF/WS/CNTs, and (**f**) schematic illustration of the photo-assisted mechanism of CF/WS/CNTs for OER under light irradiation. Adapted with permission from ref. [85]. Copyright 2022, Wiley—VCH.

Owing to the surface charge accumulation, photogenerated holes can not only directly participate in oxidation reactions, but can also promote the formation of high-valence active species to indirectly enhance the OER performance. Fu et al. confirmed that photogenerated holes on the single-component CoCr-LDH could simultaneously stimulate the production of high-valence $Co^{3+}$ active species and accelerate the charge transfer process, which are also the main reasons for the significant enhancement in electrochemical OER activity (Figure 6a,b) [30]. Similarly, Yang et al. showed that a p-n $SnS_2$/NiO heterojunction could generate more high-valence Ni ($Ni^{3+}$) on NiO as OER active sites after irradiation, which was further confirmed by XPS and electrochemical surface area (ECSA) tests after irradiation (Figure 6c,d) [45]. Such investigations provide new insights into the role of photogenerated holes in photogenerated carrier-assisted strategies. The main catalysts, experimental conditions, and activities of photogenerated carrier-assisted electrocatalysts for OER are listed in Table 2.

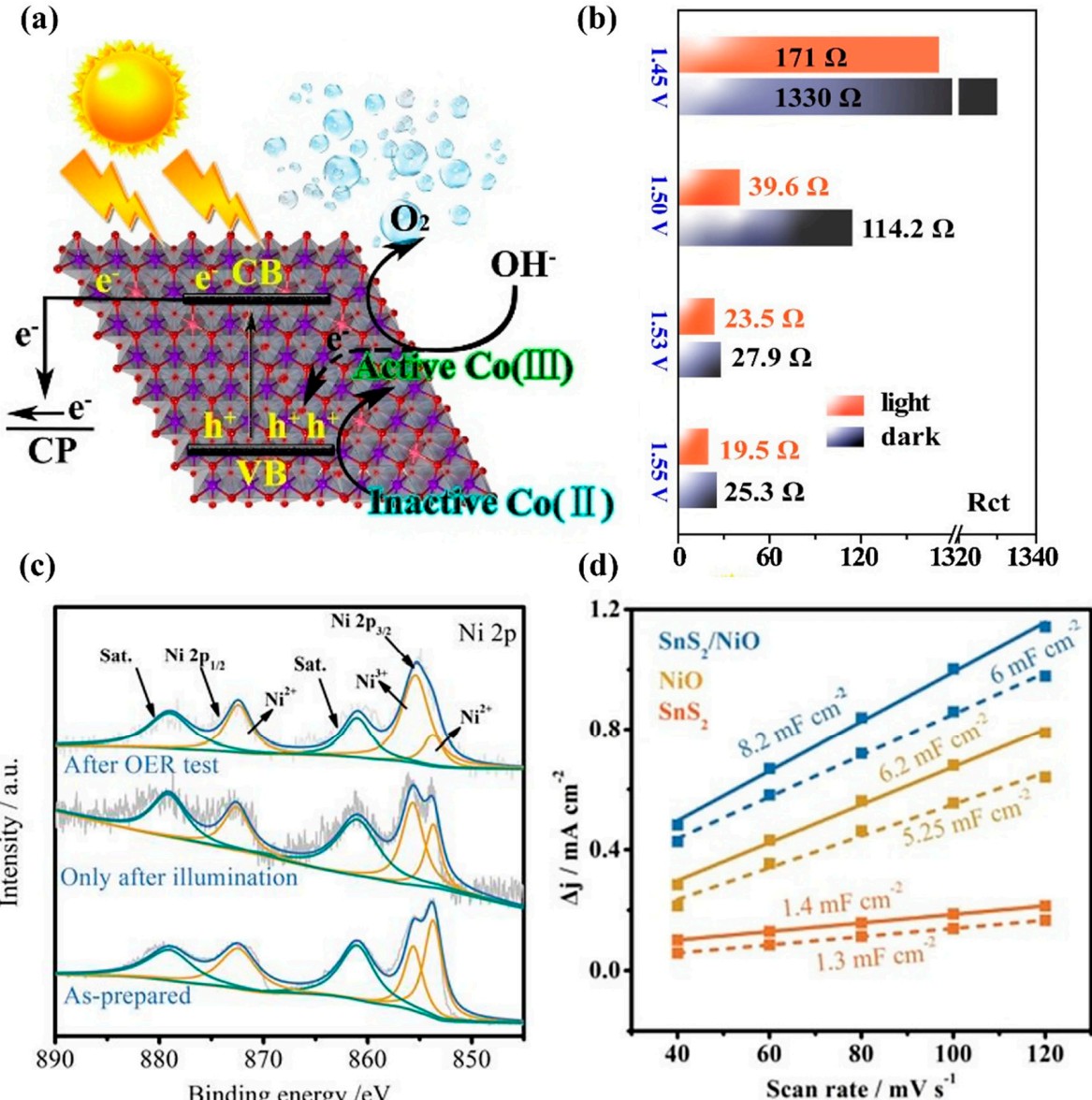

**Figure 6.** (**a**) Schematic diagram of the photogenerated hole-assisted OER mechanism for CoCr-LDH and (**b**) comparison of charge transfer resistance under different applied biases. Adapted with permission from ref. [30]. Copyright 2021, Elsevier. (**c**) Ni 2p XPS spectra of $SnS_2$/NiO and (**d**) $C_{dl}$ values of $SnS_2$/NiO under dark and light conditions. Adapted with permission from ref. [45]. Copyright 2021, Wiley—VCH.

**Table 2.** The main catalysts, experimental conditions, and activities of photogenerated carrier-assisted electrocatalysts for OER.

| Catalyst | Electrolyte | Performance Comparison, Light (Dark) | | Photocurrent Density | Ref. |
|---|---|---|---|---|---|
| | | Overpotential /mV | Tafel Slope /mV dec$^{-1}$ | | |
| IrOx@In$_2$O$_3$ | 1 M KOH | 176(190), 10 mA cm$^{-2}$<br>210(231), 50 mA cm$^{-2}$ | 47(55) | NA | [47] |
| BP QDs/MoS$_2$ | 1 M KOH | 180(330), 10 mA cm$^{-2}$ | 95(108) | NA | [78] |
| CoFe PBA/CoS$_2$ | 1 M KOH | 265(301), 10 mA cm$^{-2}$ | 59(80) | 6.5 mA cm$^{-2}$ at 1.6 V | [82] |
| FeOOH/CC | 1 M KOH | 328(352), 10 mA cm$^{-2}$ | 42(47) | NA | [84] |
| WO$_3$/SnSe$_2$/CoFe-LDH/CNTs | 1 M KOH | 224(291), 10 mA cm$^{-2}$ | 47(78) | 53 μA cm$^{-2}$ at 1.23 vs. RHE | [85] |
| CoCr-LDH | 1 M KOH | 338(360), 10 mA cm$^{-2}$ | 74(85) | 57 μA cm$^{-2}$ at $\eta$ = 70 mV | [30] |
| SnS$_2$/NiO | 1 M KOH | 310(388), 10 mA cm$^{-2}$ | 190(215) | 69.50 μAcm$^{-2}$ at 1.23 V vs. RHE | [45] |

*3.3. Photogenerated Carrier-Assisted Electrocatalysts for Overall Water Splitting*

Photogenerated carrier-assisted electrocatalysts for HER and OER usually exhibit unique physical and chemical properties to enhance the HER and OER processes, respectively. At present, non-noble metal bifunctional electrocatalysts showing high performances in both HER and OER are therefore of great significance and importance for applications in the future [86]. Therefore, the development of bifunctional photogenerated carrier-assisted electrocatalysts for overall water splitting is very attractive. The designable engineering of covalent Mo-Ni-S coupling, integrating the activities of HER and OER, shows the great application potential of dual-functional catalysts for overall water splitting [87]. Feng et al. confirmed that the interface between MoS$_2$ and Ni$_3$S$_2$ can promote the chemical adsorption of hydrogen intermediates and oxygen-containing intermediates [88]. Xu et al. reported the application of NiMoS as a photogenerated carrier-assisted electrocatalyst. Its photogenerated carrier-enhanced water splitting activity could be attributed to the transfer of photogenerated electrons to the uncoordinated active sites at the edge of Mo-S and the rapid formation of high-valence active sites for OER (Figure 7a) [89]. Yang and Fang et al. reported a series of CdS-based bifunctional photogenerated carrier-assisted electrocatalysts, including CdS/Ni$_3$S$_2$ (Figure 7b) [90], CdS/Co$_9$S$_8$/Ni$_3$S$_2$ (Figure 7f) [91], and CdS/Ni$_3$S$_2$/Ni$_x$P$_y$ [92]. CdS has a suitable visible light absorption band gap and conduction band position for water reduction, and it is commonly used as semiconductor material for photocatalytic water splitting [93]. Among them, one-dimensional CdS nanorod arrays can significantly enhance the light absorption and scattering efficiency and provide a fast electron transfer pathway for photogenerated electrons, leading to excellent optical properties [94]. The in situ growth on metal foams first ensures an enhanced electrical conductivity and mechanical stability for electrocatalytic water decomposition, and the formed Ni$_3$S$_2$/NiCoS/Ni$_x$Py at the outer end is also conducive to light absorption, the separation of photogenerated charges, and the protection of CdS from photocorrosion. DFT theoretical calculations and time-resolved photoluminescence (PL) decay spectra also demonstrated that P doping could not only improve the electrochemical adsorption of CdS/Ni$_3$S$_2$ on the water splitting intermediates, but could also prolong the lifetime of charge carriers (Figure 7c–e). The photogenerated carrier-assisted strategy totally enhanced the electrocatalytic performance of water. This is mainly attributed to the successful migration of photogenerated electron–hole pairs excited by CdS nanorods towards the anode and cathode under the external electric field and the introduction of a heterojunction structure, which can directly participate in the HER and OER processes.

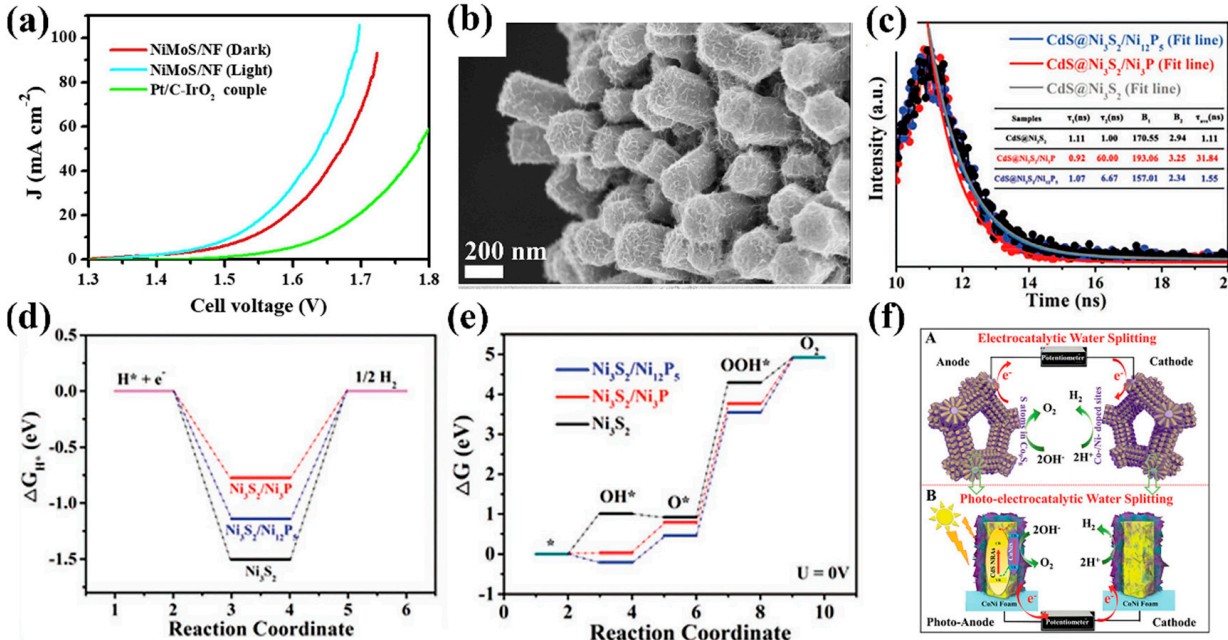

**Figure 7.** (**a**) Overall water splitting LSV curves of NiMoS in light and dark conditions. Adapted with permission from ref. [89]. Copyright 2020, Elsevier. (**b**) SEM image of CdS/Ni₃S₂. Adapted with permission from ref. [90]. Copyright 2021, Elsevier. (**c**) Time-resolved PL decay spectra of CdS@Ni₃S₂ and P-doped CdS@Ni₃S₂ and (**d**,**e**) Gibbs free energy of H* adsorption and oxygen-containing intermediate (OH*, O*, and OOH*) adsorption. Adapted with permission from ref. [92]. Copyright 2021, Wiley—VCH (**f**) Charge transport mechanism of CdS/NiCoS for electrocatalytic and photo-electrocatalytic overall water splitting. Adapted with permission from ref. [91]. Copyright 2020, Royal Society of Chemistry.

Two-dimensional transition-metal-based layered double hydroxides (LDHs) are a type of layered ionic material, and are composed of brucite-like positive charged layers formed by the coordination of trivalent metal cations with bivalent metal cations through hydroxyl substitution [95,96]. LDHs have a unique layered structure and abundant active sites and are often considered as excellent electrocatalysts [97]. Among various LDHs, NiFe LDH is regarded as one of the most effective bifunctional water splitting catalysts, and is easy to synthesize for the improvement in OER and HER activities [98]. In the past, the studies on NiFe LDH have mainly included morphology engineering, defect engineering, and the construction of layered/core–shell nanostructures [99]. An innovative strategy is to combine the photoactive components with highly active NiFe LDH in a photo-assisted electrocatalysis system to enhance its activity. Shi et al. embedded AgInZnS QDs into NiFe LDH nanoflakes by a hydrothermal treatment, and the prepared composite showed improved HER and OER properties under light irradiation. TPV tests indicated that the unique 0D/2D heterostructure could effectively extract the photogenerated charge of Ag-InZnS QDs to promote the charge transfer process (Figure 8a,e) [100]. Wang et al. prepared hydrangea-like ZnO/NiFe LDH composites with a large specific surface area by a two-step hydrothermal method. The introduction of light energy could greatly reduce the energy barrier required for the water splitting reaction. In situ Raman tests confirmed the improvement in the internal structure of the catalyst under illumination (Figure 8b,c). Combined with theoretical calculations, the electron transfer process and the intermediate adsorption were optimized, further revealing the origin of photo-enhanced activity (Figure 8f) [101]. In addition to noble metals and non-precious metals, the photogenerated carrier-assisted strategy can also be extended to metal-free dual-functional electrocatalysts. Kang et al. prepared a polyaniline (PANI)/carbon dot (CDs) composite as a metal-free bifunctional photogenerated carrier-assisted electrocatalyst [102]. PANI is a low cost, conductive polymer with a visible light response, a large π-conjugated electronic structure, fast redox characteristics,

and excellent carrier mobility [103–105]. CDs, as a kind of carbon nanomaterial, due to its environmental friendliness and excellent optical and physicochemical properties, exhibits obvious advantages in the field of catalysis [106,107]. TPV tests showed that the addition of CDs could improve the charge transfer rate and enhance the photoelectric effect of a PANI/CDs composite (Figure 8c,g). Finally, the main catalysts, experimental conditions, and activities of photogenerated carrier-assisted electrocatalysts for OER/HER (water splitting) are listed in Table 3.

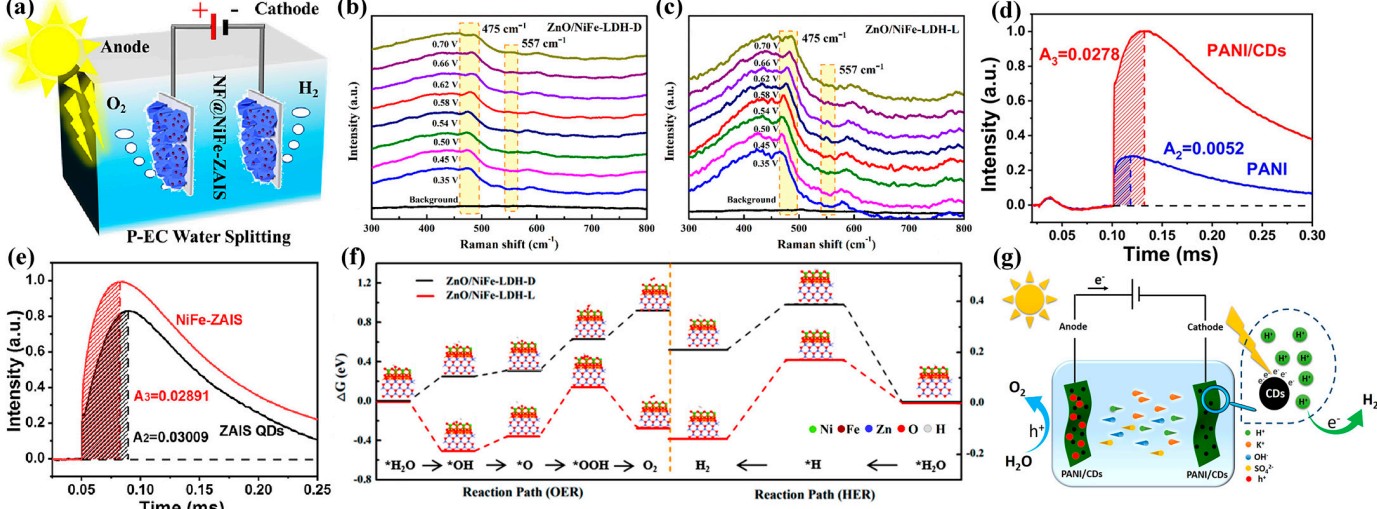

**Figure 8.** (**a**) Schematic illustration of NF@NiFe LDH-ZAIS QDs as a photogenerated carrier-assisted electrocatalytic water splitting electrolyzer and (**e**) charge extraction efficiency of NiFe LDH-ZAIS QDs. Adapted with permission from ref. [100]. Copyright 2021, American Chemical Society. In situ Raman spectra of ZnO/NiFe LDH (**b**) in the dark and (**c**) under light. (**f**) Reaction free energy of HER and OER under dark and light conditions. Adapted with permission from ref. [101]. Copyright 2022, Elsevier. (**d**) Charge extraction efficiency of PANI/CDs and (**g**) mechanism of PANI/CDs in P-EC overall water splitting. Adapted with permission from ref. [102]. Copyright 2021, American Chemical Society.

**Table 3.** The main catalysts, experimental conditions, and activities of photogenerated carrier-assisted electrocatalysts for OER/HER (water splitting).

| Catalyst | Electrolyte | Performance Comparison, Light (Dark) | | Photocurrent Density | Ref. |
|---|---|---|---|---|---|
| | | Overpotential /mV | Tafel Slope /mV dec$^{-1}$ | | |
| $MoS_2/Ni_3S_2$ | 1 M KOH | 1500(1530), 10 mA cm$^{-2}$ | NA | NA | [89] |
| $CdS/Ni_3S_2$ | 1 M KOH | NA | 75.9(85.8) | NA | [90] |
| $CdS/Co_9S_8/Ni_3S_2$ | 1 M KOH | 285(300), 10 mA cm$^{-2}$ | 87.2(91.3) | 10 mA cm$^{-2}$ at 1.56 vs. RHE | [91] |
| $CdS/Ni_3S_2/Ni_xP_y$ | 1 M KOH | NA | 103(108.7) | NA | [92] |
| AgInZnS QDs/NiFe-LDH | 1 M KOH | 1620(1670), 10 mA cm$^{-2}$ | 105(111.7), 54.6 (71.6) | NA | [100] |
| ZnO/NiFe-LDH | 1 M KOH | 1630(1730), 10 mA cm$^{-2}$ | 87.03(219.09), 67.28(201.47) | NA | [101] |
| PANI/CDs | 0.5 M K$_2$SO$_4$ | Decrease by 150 mV @30 mA cm$^{-2}$/65 mV @20 mA cm$^{-2}$ | 192(283) | 5.12 mA cm$^{-2}$ at 2.0 V | [102] |

## 4. Conclusions and Prospects

In summary, we reviewed the latest progress in photoelectric integrated catalysts using PCAS to enhance the activity of HER, OER, and overall water splitting. In the photogenerated carrier-assisted strategy, electron–hole pairs excited by the photoactive components in photoelectric integrated catalysts under illumination can directly participate in redox reactions. Note that the photogenerated holes can additionally promote the formation of high-valence active species for OER. This simple and efficient strategy for enhancing the activity has become more attractive in applications for HER, OER, and overall water splitting. Based on the feature of HER, OER, and water splitting, the structure and composition of the corresponding photogenerated carrier-assisted electrocatalysts were rationally designed to suit all kinds of HER, OER, and overall water splitting. Three aspects of HER, OER, and water splitting were especially reviewed for the development of a single-component photoelectric response and the coupling of photoactive components and electrocatalysts. Among them, to accelerate the separation of charge carriers, the construction of heterogeneous interfaces and the modification of highly conductive species are the most efficient approaches. Although some aspects have made great progress, the overall amount of research in photogenerated carrier-assisted electrocatalysis is still relatively low. There is a huge research gap which needs to be filled and explored to improve the efficiency of photoelectric integrated catalysts in HER, OER, and water splitting. Based on the above-mentioned results and the corresponding issues, some pivotal challenges that need to be confronted and the prospects of improving photogenerated carrier-assisted electrolysis are addressed in the following.

Integrating photocatalytic and electrocatalytic materials into one electrode is still a challenge. Although single-component photoelectric integrated catalysts have uncomplicated compositions and strong photoelectric coupling interfaces, they face the problems of a low electrocatalytic activity or a poor photo-assisting effect. It is worth noting that some layered MOF derivatives and transition metal oxides with abundant photoelectric properties show promising prospects as single-component photoelectric integrated catalysts. Semiconductor materials are not ideal electrocatalysts due to their low conductivity. When the photoactive component and the highly active electrocatalyst are synergistically coupled together to enhance the photo-assisting ability and electrocatalytic activity, the introduction of semiconductors may affect the overall performance of the photogenerated carrier-assisted electrocatalyst. Coordinating the photoelectric characteristics of semiconductor materials and deeply understanding the electrochemical behavior of semiconductors are conducive to their application in photogenerated carrier-assisted electrocatalysis systems. Recent investigations have discovered the abnormal phenomenon of carrier concentration on the semiconductor surface, defined as the self-gating effect. This effect can effectively adjust the surface conductivity of semiconductor electrocatalysts, and therefore succinctly explains why ultra-thin semiconductors are highly efficient electrocatalysts [108]. Other studies have revealed that charge transfer processes can occur in some active crystal surfaces and atomic-level active sites of semiconductors. For example, $MoS_2$ edge sites have been proven to be an HER active center. In addition to the above-mentioned strategy, n-type and p-type semiconductors have also been proven to be beneficial to cathodic reduction reactions and anodic reactions, respectively. All these strategies provide effective insights into semiconductor electrochemical applications. The rational design of the structure and interfacial properties of photoactive and electroactive components can transform components from inert to active.

The recombination of photogenerated carriers seriously influences the efficiency of photogenerated carrier-assisted electrocatalysis. The effective injection of photogenerated carriers is key to the photo-enhanced activity, but the inevitable recombination in the migration process greatly affects the utilization efficiency of carriers. At the same time, the accumulation of photogenerated carriers on the surface may lead to photoelectrode corrosion, seriously affecting the long-term stability of photogenerated carrier-assisted electrocatalysts. The rational design of photo-assisted electrocatalysts with a high carrier

mobility is crucial for the synergetic use of light energy to enhance energy conversion. In traditional photo(electro)catalysis, many studies have focused on the adjustment of the charge transfer kinetics of photoelectrodes, from which we can gain inspiration. The construction of a heterojunction can form a built-in electric field (BIEF) at the contact interface and promote charge separation, which is one of the most important goals. Different types of heterojunction structures will provide different charge transfer mechanisms to adjust the electron density. In electrocatalysis, a BIEF has also been proven to be crucial in the adsorption/desorption behavior of reactants and key intermediates. Therefore, it is promising to synergize the rapid charge transfer process and the optimized electrochemical behavior of heterojunctions in a photogenerated carrier-assisted electrocatalysis system. In addition, element doping and defect engineering can adjust the band structure of photoactive components to produce more photo-induced carriers and provide more charge capture centers and adsorption sites for promoting charge separation, which can be used as other strategies in photogenerated carrier-assisted electrocatalytic design.

The practical applications of photogenerated carrier-assisted electrocatalysis still need to be further developed. On the one hand, the economic issues between the input light energy and the enhanced output activity need to be considered. Although the introduction of light energy can reduce the overpotential of the water decomposition process and compensate for power consumption, additional light energy will also increase the economic cost of the energy input. Therefore, to reduce the cost of optical assistant strategies, it is important to enhance the efficiency of light utilization and optimize the intrinsic electrocatalytic activity of photogenerated carrier-assisted electrocatalysts in order to reduce the overpotential of redox reactions. The rational design of photogenerated carrier-assisted electrocatalysts with an optimized electronic structure to promote the adsorption of electrochemical intermediates and the separation of photogenerated carriers is very promising and conducive to the practical application of photogenerated carrier-assisted electrocatalysis. On the other hand, the design of a large-scale photogenerated carrier-assisted electrolytic cell is essential, but the research on photoelectric devices is still relatively scarce. Industrial-scale electrolytic water splitting technology has been relatively well-researched; embedding light-transmitting windows makes the practical application of photogenerated carrier-assisted technology possible.

**Author Contributions:** Conceptualization, Y.Z. and Y.L.; investigation, X.L. and X.Z.; writing—original draft preparation, X.L.; writing—review and editing, X.L., X.Z. and Y.L.; supervision, Y.Z. and Y.L. All authors have read and agreed to the published version of the manuscript.

**Funding:** This research was supported by the Major Research and Development Project of Central Government Guides Local Science and Technology Development Professional Technology Innovation Platform (no. 2019ZY-CXPT-08), the Regional Innovation Capability Leading Program of Shaanxi (2022QFY07-03 and 2022QFY07-06), the Key R&D program of Shaanxi Province (2021GY-166), and the Natural Science Program of the Education Department of Shaanxi Province (20JS155).

**Data Availability Statement:** No further data are available.

**Conflicts of Interest:** The authors declare no conflict of interest.

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
