# Peer review of "Photogenerated Carrier-Assisted Electrocatalysts for Efficient Water Splitting"

_catalysts, doi:10.3390/catal13040712_

Round 1
Author Response
Please see the attache file: Authors' response for reviewer 1

Reviewer 2 Report
In this review, authors introduced the innovative strategy of photogenerated carrier-assisted electrocatalysts for water splitting, systematically and comprehensively summarized the photoelectric integrated catalysts’ structure composition, photoelectric properties and their application in HER, OER and overall water splitting. The overall work is satisfactory and has certain guiding significance. Therefore, I recommend its publication in Catalysts after minor revision according to the suggestions below.
1. Partial characters format in the page 3, 14 is not standardized.
2. In Figure 1, some of the cited pictures cannot directly display their photo-assisted applications.
3. The subtitles of section 3 should correspond to the title.
4. The consideration of economic issue in conclusion section is not very appropriate.
5. The description of ZnO/NiFe LDH, such as in situ Raman and theoretical calculations, is inconsistent with the results provided in Figure 7.
6. Some relevant works can be cited and discussed in the manuscript to attract more readerships (e.g. Nano Res. 2022, 15, 7026-7033; Adv. Funct. Mater. 2018, 28, 1802685; J. Mater. Sci. Technol. 2022, 104, 155-162; Nanoscale 2021, 13, 17989-18009; J. Mater. Chem. A 2020, 8, 24307-24352).
Author Response
Please see the attached file: Authors' response for Reviewer 2

Reviewer 3 Report
This review aims at summarizing the development of photogenerated carrier-assisted strategy (PCAS) for enhancing the performance of HER, OER and water splitting. This review could provide systematic background knowledge and give some insights to researchers working in this field. Before considering for publication, there are some major concerns for authors to address.
1. There are many repetitive expressions. For instance, “In this review, we will first focus on the latest progress of photogenerated carrier assisted electrocatalysis, understandably elucidate its possible mechanism for the enhancement of performance by summarizing the reaction mechanism of HER and OER. Then, recent advances in electrocatalysts for assisted enhanced HER, OER, and total water splitting by photogenerated carrier strategies are introduced, and their design and fundamental principle are briefly described.” Please check the whole article and refine the language to make it more readable.
2. The resolution of Fig 7e should be improved.
3. As an advanced review, authors are suggested to discuss the development of single-component photoelectric response combination, the coupling of photoactive components and electrocatalysts instead of just listing the existing experimental results.
4. The article states that “Integrating photocatalytic and electrocatalytic materials into one electrode is still a challenge.” What are the possible strategies for overcoming this difficulty?
Author Response
Please see the attached file: Authors' response for Reviewer 3.

Reviewer 4 Report
In this manuscript, Li et al. provided a very timely review on a very emerging topic, which is Photogenerated Carrier-Assisted Electrocatalysts for Efficient Water Splitting. This research direction is gaining increasing attention but a comprehensive review summarising the recent achievements was not available in the literature. The manuscript focused on the fundamental understanding and the recent developments in HER/OER and overall water splitting, while also presenting some of their own perspectives in this research direction. Overall, this review work is a good fit for the journal Catalysts. I would in general support publication. However, some minor technical issues need to be resolved before the possible acceptance of the manuscript. Please see below for more detail.
1. The review summarizes catalyst materials in both alkaline and acidic electrolytes, while the mechanism study (in section “2. Fundamental principle of photo-assisted electrocatalysis strategies”) is introduced based on the alkaline condition. Please specify in section 2 that the discussion was made in alkaline electrolyte.
2. To appeal to a broader readership, very recent works on water electrolysis, HER and OER catalysts are suggested to be referenced in the Introduction section (e.g., Energy Technology, 2022, 10, 2200573; Materials Reports: Energy, 2022, DOI: 10.1016/j.matre.2022.100144).
3. In the Conclusion section, the authors mentioned “reasonable design of photoelectrocatalysts”. However, the manuscript actually focuses on design of electrocatalysts assisted by photogenerated carrier. Please double check to address any inconsistency if present.
4. Indeed, “the construction of heterojunction can … modify the properties of active sites.” Recent works on heterostructured OER catalysts can be referred to (e.g., Small, 2021, 17, 2101573).
5. For Table 1-3, what are the numbers in the parenthesis, e.g., for Tafel slope and for performance comparison?
6. Keywords of the article should be provided.
Author Response
Please see the attached file: Authors' response for Reviewer 4.

Round 2
Reviewer 3 Report
I suggest the manuscript to be accepted in the present form.